# Travelers’ Attitudes, Behaviors, and Practices on the Prevention of Infectious Diseases: A Study for Non-European Destinations

**DOI:** 10.3390/ijerph18063110

**Published:** 2021-03-18

**Authors:** Angela Bechini, Patrizio Zanobini, Beatrice Zanella, Leonardo Ancillotti, Andrea Moscadelli, Paolo Bonanni, Sara Boccalini

**Affiliations:** 1Department of Health Sciences, University of Florence, 50134 Florence, Italy; patrizio.zanobini@unifi.it (P.Z.); beatrice.zanella@unifi.it (B.Z.); leonardoancillotti24@gmail.com (L.A.); paolo.bonanni@unifi.it (P.B.); sara.boccalini@unifi.it (S.B.); 2Medical Specialization School of Hygiene and Preventive Medicine, University of Florence, 50134 Florence, Italy; andrea.moscadelli@unifi.it

**Keywords:** travel, traveler, behavior, attitude, travel risk, non-European countries, infectious diseases, vaccines, vaccination, online survey

## Abstract

Background: The aim of this study was to improve our comprehension of the attitudes, behaviors, and practices related to the health risks of travel to non-European destinations. Methods: This cross-sectional study utilized an online questionnaire disseminated through social networks. Results: About 79% of the respondents reported that they informed themselves or would inform themselves about travel-related health risks before departure. The most used sources of information were the Internet (52%) and health professionals (42%). Older age groups were positively associated with seeking pretravel health information (OR = 2.44, CI 95%: 1.18–5.01, in the age group 25–34 years and OR = 14.6, CI 95%: 1.77–119.50, in subjects over 65 years). Travelers visiting friends and relatives (VFRs) were less likely to seek health information (OR = 0.49; CI 95%: 0.26–0.95). About 13.9% of participants had doubts about the practice of vaccination. Those who sought information on social media had a higher probability of refusing vaccination (OR = 3.24; CI 95%: 1.02–10.19). Conclusions: The data demonstrate that VFRs and the younger population are less informed about travel risks compared with other travelers. This study also revealed the importance that digital information assumes for travelers. Therefore, decisive efforts should be made to ensure that travelers can find correct and reliable information on the Web, particularly on social media.

## 1. Introduction

According to the World Tourism Organization, the number of international travelers increased by 6% in 2018, reaching 1.4 billion arrivals all over the world. The destinations with the most significant increase in arrivals were the Middle East and Africa, with increases of 10% and 7%, respectively. Growth in the number of visitors to Asia and Europe was comparable to the world average (+6%) [1].

Considering the large number of international travelers, it is of great interest to be aware of possible health risks. Up to one-third of travelers contract an illness during their trip or immediately after their return. Moreover, one of the main causes of death among travelers is trauma followed by infectious diseases [2]. In a 2006 study carried out at a travel medical center in Florence by Laverone et al. [3], anonymous questionnaires were administered to travelers returning from international destinations. The questionnaire assessed travelers’ vaccination cycles, started before departure, and found that only 55% of 1237 participants took antimalarial prophylaxis, and 88% of these followed the indications given during counseling prior to departure. In a cross-sectional study [4] conducted from May to September 2018, a random group of individuals who visited a travel agency located in the Campania region were interviewed regarding travel risks and the preventive measures needed to avoid them. Data were collected from 422 participants and revealed a low level of knowledge of travel-related infectious diseases and poor adherence to preventive measures before departure. Other recent studies [5,6] noted that international travelers are at risk due to poor compliance with vaccination. The major reason travelers rejected all vaccines was poor knowledge of the severity of vaccine-associated diseases [6].

Data from the literature indicate the need to understand the habits and behaviors of international travelers, and based on current knowledge, studies that include online-administered investigations are absent. Moreover, there is a need to investigate if different sources of Web-based information may influence travelers’ behaviors. Considering the growing use of the Internet as a source of health information, online surveys can be useful to properly address poor traveler compliance with preventive measures. Moreover, online surveys can reach a wider population (ideally, all Internet users), and, in particular, people who do not attend places for the gathering of information related to traveling (travel agencies, traveler medical centers, embassies, etc.) or airports.

The aim of this study was to improve knowledge of the attitudes, behaviors, and practices of travelers to non-European destinations through an online questionnaire disseminated by means of social media.

## 2. Materials and Methods

This cross-sectional study involved the use of an online ad hoc questionnaire. The questionnaire was developed by a panel of experts and was based on the structure of questionnaires administered in similar studies [3,4]. A pilot study was carried out with a small sample of volunteers (30 subjects) to evaluate participation in the project, correct interpretations of the questions by all responders, and provide correspondence between the written answers given by responders and their replies in a parallel interview conducted by one of the authors.

The questionnaire was disseminated through social networks, primarily Facebook. An online link to the anonymous questionnaire (using Google Forms) was included in a post published on Facebook. Participants could share this link with their contacts via social platforms (other than Facebook) or instant messaging applications.

The survey was addressed to Internet or social media users who were over 18 years old and living in Italy. The only additional eligibility criterion was to provide consent to the study.

The anonymous questionnaire used is included in Appendix A.

### Statistical Analysis

Socio-demographic characteristics included in this analysis were:
Sex (male; female);Age group (18–24; 25–34; 35–44; 45–54; 55–64; 65+)Education (elementary school diploma; secondary school diploma; high school diploma; university degree; postgraduate qualification);Occupation (houseworker; unemployed; worker; retired; student);Citizenship (Italian; foreign).

Traveling behaviors and practices investigated were:
Reason for travel (study, leisure/tourism; work; voluntary work; visiting friends and relatives);How far in advance the trip was planned (<1 week; >1 week and <2 weeks; >2 weeks and <1 month; >1 month);Seeking pretravel health information (yes; no; do not know);Information sources (health professional; family or friends; the Internet; embassies or consulates; vaccination or traveler medical clinic; books; apps; travel agency; pharmacist; social networks; other);Refusing vaccinations before a trip (yes; no);Attention given to food (fruits and vegetables; raw seafood) and drink (tap water; bottled water; ice cubes) during travel (yes; no; do not know).

We deleted all observations that had a missing age value or that reported a residence other than Italy. In order to analyze the main characteristics of the participants, descriptive statistics were used, including the frequency and percentage of variables related to the demographic characteristics: sex, age group, education, traveling behaviors, and pretravel health practices (seeking information and attitude toward vaccination). Bivariate analyses were performed, using chi-square tests, to identify the significant factors associated with seeking health information before traveling and attitude toward vaccinations. A *p*-value less than or equal to 0.05 was considered statistically significant. All significant variables from the bivariate analysis were included in the logistic multivariate models. Independent variables were selected using a backward elimination procedure, retaining only age, sex, and those variables with *p* ≤ 0.05.

Statistical analyses were performed using Stata 15.0 (STATA Corp LLC, 4905 Lakeway Drive, College Station, TX, USA).

## 3. Results

### 3.1. Socio-Demographic Characteristics

The survey was available during the second week of September 2019, and 418 questionnaires were collected. Most participants were Italian (98%), female (66%), workers (75%), aged between 25 and 44 years (59%), and had a university degree or a postgraduate qualification (68%) (Table 1).

### 3.2. Travelers’ Attitudes, Behaviors, and Practices

About 79% (*n* = 334) of participants stated they traveled outside the European Union, and 93% (*n* = 385) intended to travel outside the European Union in the future.

Most participants traveled for leisure/tourism (94.5%) and planned their trip more than one month before departure (79.9%) (Table 2).

Participants were asked whether they generally informed themselves or would inform themselves before departure about travel-related health risks; 79.4% reported doing so. Among those who informed themselves or would inform themselves before traveling, the majority (52.4%, *n* = 257) sought information on the Internet, followed by doctors or health professionals (42.4%, *n* = 208). About 80% of participants used two or more sources of information.

Participants were asked if they paid attention to certain foods when they traveled outside of the European Union. About 15.5% were not wary of raw fruits and vegetables, 21.7% of ice cubes, 3.6% of tap water, 70.9% of bottled water, and 15.5% of raw seafood (Table 3).

### 3.3. Attitudes toward Vaccination

Participants were asked if they refused or would refuse vaccinations before a trip; 86.1% underwent or would undergo the recommended vaccinations, while 13.9% had doubts about vaccination or were against it. Among those, the most common reason for not getting vaccinated was the fear of possible side effects (46.6%) (Table 4).

### 3.4. Predictors of Seeking Health Information before Travel and Vaccination Attitudes

In the original multivariate model for seeking information before travel, the following variables were included: age group, sex, reason for travel (leisure/tourism; visiting friends and relatives; study).

Table 5 shows the results from the final multivariate logistic regression analysis of factors that influenced pretravel information seeking. The final model includes the variables that maintained a statistically significant association with the outcome variable applying the backward stepwise procedure. Age was positively associated with seeking information. On the contrary, those who traveled to visit friends and relatives (VFRs) had a lower probability of seeking health information (OR = 0.49; CI 95%: 0.26–0.95).

In the original multivariate model for refusing vaccination, the following variables were included: education, source of information (i.e., social networks), and number of information sources.

Table 6 shows the predictors of refusing vaccination before a trip after applying the backward stepwise procedure. Those who sought information on social media had a higher probability of refusing vaccination before a trip (OR = 3.24; CI 95%: 1.02–10.19).

## 4. Discussion

The aim of our study was to improve comprehension of the attitudes, behaviors, and practices relating to health risks of travelers to non-European destinations through an online questionnaire. The results of the study showed that the majority of the population had traveled or intended to travel (93%) internationally, confirming the importance of travelers’ medicine and the importance of counseling before embarking on a journey.

About 79% of the respondents informed themselves or would inform themselves about travel-related health risks before departure, mostly through the Internet (52.4%) and health professionals (42.4%). Age was positively associated with seeking pretravel health information (OR = 2.44, CI 95%: 1.18–5.01, in the age group 25–34 years and OR=14.6, CI 95%: 1.77–119.50, in subjects over 65 years). On the other hand, those who traveled to visit friends and relatives (VFRs) had a lower probability of seeking out health information (OR = 0.46; CI 95%: 0.26–0.95). Approximately 85% of participants affirmed they had undergone/would undergo the recommended vaccinations. Those who sought information on social media had a higher probability of refusing vaccination (OR = 3.24; CI 95%: 1.02–10.19).

Tafuri et al. [7] demonstrated the effectiveness of counseling before travel. However, only 36% of those who left for international destinations sought information on health risks before their journey. A European study showed that, due to a lack of understanding of potential travel risks, many individuals traveling within the region wrongly believed that no medical advice or vaccination was required [8]. Our results highlighted a high percentage of participants who searched for information before traveling, allowing us to assume that most of our study population may be well-prepared regarding travel risks, and may pay more attention to practices such as vaccination that need to be followed before traveling abroad, compared with other studies [4,9]. These data are confirmed by the high number of participants that seemed to be particularly aware of the risks they may face in drinking contaminated water or consuming raw fruits or vegetables. Foodborne and waterborne infections are a ubiquitous risk that travelers should be aware of [10]. Often, VFR travelers are less likely to seek pretravel health advice compared with other travelers [11,12,13,14], with up to 30% of VFR travelers seeking pretravel advice compared with up to 60% of other travelers [15,16]. While we also found that VFRs were less likely to seek pretravel information, the percentage of VFRs seeking pretravel information (67%) was higher than that presented elsewhere [17].

These encouraging findings related to seeking information before travel may be explained by the higher education level of our sample compared with the general population (66% vs. 14% university graduates, respectively) [18].

Concerning the sources of information, our results are in line with the literature. The Internet has become the preferred information source for approximately 50% of travelers [19].

The likely adherence to recommended vaccination among the interviewed people (about 86%) also seems to be in line with the latest report on vaccine confidence for European countries, published in 2018, which showed that confidence in vaccination has increased in Italy since 2015 [20]. Moreover, according to other surveys, 80–90% of Italian parents accepted vaccinations for their children [21,22].

The major cause identified for refusing all vaccines is poor knowledge of the severity of vaccine-associated diseases [6]. According to a review published in 2019 [23], there is a significant positive association between those who consult health care professionals for information and adherence to recommended vaccinations, but in our regression model this association was not significant. Indeed, those who seek information on social networks have a higher probability of refusing vaccination compared with those who do not. This is of particular interest because we also considered the Internet and apps as sources of information in our analysis, but neither of the two were significantly associated with the refusal to get vaccinated. As suggested by Wilson et al. [24], multiple studies from the early 2000s reported that a large proportion of vaccine-related content on popular social media sites are antivaccination messages.

Healthcare workers need to know about and use new ways of communication to disseminate high-quality research and to fight misinformation and the phenomenon of vaccine hesitancy [25]. Furthermore, interventions should aim to produce and disseminate evidence-based, solid, comprehensive, understandable, and updated information about vaccines on social media to counterbalance misleading and erroneous information and thereby increase vaccination confidence [26]. In particular, younger participants seem to be less inclined to seek health advice. This could suggest the need to plan targeted interventions in order to overcome this issue; for example, it could be useful to raise awareness about travel risks through the use of social media or apps. Moreover, the school setting could be a possible place to carry out interactive lessons in order to increase knowledge among adolescents or young adults before they are exposed to travel risks (i.e., the first time traveling alone or without parents during study abroad experiences). 

Our study has some limitations. Since the participants came from all over the national Italian territory, findings are not geographically limited as in other studies conducted directly in a small territory. This lack of location-specific data prevents making considerations on specific areas, as the sample is dispersed across national soil.

In order to keep the number of applications limited and to make it easier to complete the online questionnaire, we did not define or identify non-European destinations. These destinations could include countries with food, water quality, hygiene, and disease profiles similar to those in Europe, or they could include developing countries that would probably be perceived as riskier. Moreover, questions regarding pregnancy status or travelers with comorbidities were not included. Lastly, we did not consider the risk of acquiring multidrug resistant organism (MDRO) infections while traveling abroad [27,28]. This can occur even if travelers do not have contact with the healthcare system of the country.

Finally, the use of an online survey allowed us to easily reach a large number of people in a short time; however, this may result in sampling bias due to exclusion of the non-digitalized population and those who rarely use online social media. It is notable that adults between the ages of 18 and 35 were over-represented in this survey sample, while older adults ages 55–65+ were under-represented. Furthermore, participation in the survey was voluntary. Antivaxxers are probably more likely to want to make their opinions known, compared with those who have neutral or positive attitudes to vaccinations. However, the percentage of responders refusing vaccination seemed to be in line with the most recent report [20].

Finally, the responses may be subject to self-reporting bias and the tendency to report socially desirable responses.

## 5. Conclusions

In conclusion, the data show VFRs and the younger population are less informed about travel risks compared with other groups, underlying their higher risk profiles and indicating the need to give these groups more specific attention. Moreover, as far as information sources are concerned, this study revealed the importance that digital information assumes for travelers. Therefore, decisive efforts should be made to ensure that travelers can find correct and reliable information on the Web, particularly on social media. As a future perspective, it would be interesting to assess how travelers’ attitudes, behaviors, and practices may vary for different destinations by comparing those of travelers to countries with hygiene standards similar to Europe with those of travelers to countries with poor hygiene and limited access to safe drinking water and sanitation services.

## Figures and Tables

**Table 1 ijerph-18-03110-t001:** Socio-demographic characteristics of the study population.

Sociodemographic Characteristics	*n*	%	Sociodemographic Characteristics	*n*	%
**Sex**			**Education**		
Female	274	65.6	Secondary school education	5	1.2
Male	144	34.4	High school diploma	126	30.4
**Age (years)**			University degree	202	48.8
18–24	42	10.5	Postgraduate qualification	81	19.6
25–34	155	37.1	Missing	4	
35–44	83	19.9	**Profession**		
45–54	68	16.3	Houseworkers	9	2.1
55–64	49	11.7	Unemployed	11	2.6
>64	21	5	Workers	314	75.5
			Retired	20	4.81
**Citizenship**			Students	62	14.9
Italian	408	98.1	Missing	2	
Foreign	8	1.9			
Missing	2				

**Table 2 ijerph-18-03110-t002:** Travelers’ attitudes and behaviors.

Answers	*n*	%		*n*	%
**Reason for travel**			**Information sources**		
Study	43	10.3	Health professional	185	55.7
Leisure/tourism	395	94.5	Family or friends	59	17.8
Work	81	19.4	Internet	215	64.8
Voluntary work	29	6.9	Embassies or consulates	76	22.9
Visit friends and relatives	55	13.2	Other	14	4.8
**How far in advance** **the trip was planned**			Vaccination clinics or travelers’ medical clinics	108	32.5
1 week	16	3.8	Books	41	12.3
>1 week <2 weeks	31	7.4	Apps	55	16.6
>2 weeks <1 month	95	22.8	Travel agency	63	18.9
>1 month	333	79.9	Pharmacist	25	7.5
**Seeking pretravel information**			Social networks	16	4.8
Yes	387	79.0	**Number of information sources used**		
No	69	14.1	1	69	20.8
Do not know	34	6.9	2	87	26.2
			3 or more	176	54

**Table 3 ijerph-18-03110-t003:** Food and drink risk perception among participants.

Answers	Bottled Water		Ice Cubes		Raw Seafood		Raw Fruits and Vegetables		Tap Water	
	*n*	%	*n*	%	*n*	%	*n*	%	*n*	%
No	290	70.9	90	21.7	64	15.5	62	14.9	15	3.6
Do not know	9	2.2	22	5.3	25	6.1	18	4.3	6	1.5
Yes	110	26.9	302	72.9	324	78.4	337	80.8	394	94.9

**Table 4 ijerph-18-03110-t004:** Attitudes toward vaccinations.

Questions and Answers	*n*	%
**Refuse vaccination**		
No	358	86.1
Yes/Do not know	58	13.9
**Reason to not get vaccinated**		
Fear of side effects	27	46.6
Not useful	10	17.2
Do not have time	9	15.5
Cost	4	6.9
Difficulties in gaining access to vaccination clinic	4	6.9
Other	24	41.4

**Table 5 ijerph-18-03110-t005:** Multivariate logistic regression analysis: predictors of seeking information before traveling.

Predictors	Seeking Health Information, *n* (%)		Adjusted OR	CI 95%
	No	Yes		
**Time dedicated to the travel plan**				
>2 weeks	16 (32.65)	33 (67.35)	Reference group	
up to 2 weeks	87 (19.86)	351 (80.14)	1.72	0.86–3.45
**Age group**				
18–24	18 (42.86)	24 (57.14)	Reference group	
25–34	35 (22.58)	120 (77.42)	2.44	1.18–5.01
35–44	16 (19.28)	67 (80.72)	3.0	1.31–6.92
45–54	9 (13.24	59 (86.76)	4.0	1.56–10.37
55–64	7 (14.29)	42 (85.71)	4.1	1.48–11.41
65+	1 (4.76)	20 (95.24)	14.6	1.77–119.50
Sex				
Male	38 (26.39)	106 (73.61)	0.61	0.37–1.02
Female	48 (17.52)	226 (82.48)	Reference group	
**Reason for travel**				
Visiting friends and relatives (VFR)	18 (32.73)	37 (67.27)	0.49	0.26–0.95
Not VFR	68 (18.73)	295 (81.27)	Reference group	

**Table 6 ijerph-18-03110-t006:** Multivariate logistic regression analysis: predictors of refusing pretravel vaccinations.

Predictors	Refused Vaccinations, *n* (%)		Adjusted OR	95% CI
	Yes	No		
**Age group**				
18–24	8 (19.05)	34 (80.95)	Reference group	
25–34	17 (10.97)	138 (89.03)	1.40	0.29–6.69
35–44	16 (19.75)	65 (80.25)	3.33	0.69–16.01
45–54	11 (16.18)	57 (83.82)	2.37	0.47–12.02
55–64	5 (10.20)	44 (89.80)	0.94	0.14–6.17
65+	1 (4.76)	20 (95.24)	0.57	0.47–6.89
Sex				
Male	20 (13.99)	123 (86.01)	1.12	0.56–2.24
Female	38 (13.92)	235 (86.08)	Reference group	
**Seeking information** **on social media**				
No	40 (12.70)	275 (87.30)	Reference group	
Yes	5 (31.25)	11 (68.75)	3.24	1.02–10.19

## Data Availability

Data sharing not applicable. Data were collected and managed in aggregated form according to European Union Regulation 2016/679 of the European Parliament and the Italian Legislative Decree 2018/101.

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
