# Peer review of "Travelers’ Attitudes, Behaviors, and Practices on the Prevention of Infectious Diseases: A Study for Non-European Destinations"

_ijerph, 2021, doi:10.3390/ijerph18063110_

Round 1

Reviewer 1 Report

Authors have sufficiently addressed the review comments

Reviewer 2 Report

My previous comments have all been satisfactorily addressed and the manuscript is improved.

Reviewer 3 Report

L37: the authors must note that the health risks may encompass communicable and non-communicable causes of morbidity as well!

L54-55: the reason for performing the study is that there is limited studies available that were internet based? that is quite a weak explanation. please conceptualize your aims better!

L57-59: yes, internet-based studies may reach a wider audience, but still, that is no research gap.

L65-70: please provide more data on the questionnaire-development, because this highly affects the quality of the results.

L117: and people who had an university degree…

The authors discuss their findings appropriately in the context of International data. I would include one additional topic, namely the risk of acquiring the carriage of a multidrug resistant organism (MDRO) e.g., in the gut of individuals travelling abroad, even if they do not have contact with the healthcare system of the country.

Please consider including the following reference:

https://www.mdpi.com/2079-6382/10/3/268

Limitations: appropriately described, although a separate section should be dedicated to this.

Author Response

This manuscript is a resubmission of an earlier submission. The following is a list of the peer review reports and author responses from that submission.

Round 1

Reviewer 1 Report

Line 37: Please cite authors and not their affiliation.

Line 60: replace ‘spreading’ with ‘disseminating’.

Authors have not provided a strong rationale for this study. How is the aim stated at the end of the introduction section different from the evidence presented from the literature reviewed? It is unclear what online dissemination of a questionnaire will add to the quality of evidence in this area, this needs to be unpacked.

Biases, such as sampling bias, due to the method of dissemination of the questionnaire were not considered. What are the limitations of this in terms of representativeness and in capturing the experiences of specific demographic groups of travellers such as the elderly who might not be as much engaged in online platforms and social media as the younger demography?

It is unclear how eligibility criteria on age and residency were applied. How did you ensure all participants met the eligibility criteria?

The methods section is missing many key details. There needs to be a subsection on the data to discuss the validity and reliability of the data. Was the questionnaire validated through processes such as pilot, test-retest etc? No detail on missingness and how authors have handled missing values, this is a key feature of questionnaire data. The variables (dependent, independent, covariates etc) included in the analyses need to be described with their measurement level identified.

Overall, the analysis is too descriptive. The inferential statistics conducted was vaguely reported. Was this a regression model? If so authors should report this and model specification used. Why was a multivariable regression model not considered?

It is unclear whether the questions on travel history and destination were asked for a specific period. If not have the authors considered recall bias?

In describing the number of participants who inform themselves by sources, authors should indicate whether participants indicated multiple sources.

Too many descriptive stats in the results text, most of these are already reported in tables 2-4. Authors should report the important descriptive and refer to tables 2-4 for more descriptive results.

Please include a multivariable regression model adjusting for age, gender, travel reason, etc in modelling the predictors of seeking pre-travel information and vaccination refusal. Also, issues such as multicollinearity between covariates, if experienced, should be discussed in the introduction section.

The discussion section needs restructuring using the traditional structure of a discussion paper. The discussion should be restructured around the following areas:

  • Reiterate the research problem and main findings.
  • Explain the meaning and relevance of the main findings
  • Compare findings to similar studies
  • Consider alternative explanations of the findings
  • Policy, practice and research implications of the findings.
  • Acknowledge limitations

Authors have not demonstrated the policy, practice or research implications of the study and its findings. They ‘so what’ question has not been answered in this paper.

Reviewer 2 Report

This an interesting exercise in using social media as a vehicle to survey Italian potential travellers on health matters related to travel outside Europe. 

A few comments and suggestions:

Methods

The process of inserting the questionnaire into social media and catalysing its onward distribution is not described; this could be of interest to others considering similar surveys. Presumably there was some message encouraging participation and sharing of the questionnaire.

Other limitations

  1. Non-European destinations were not defined or identified in any way. These destinations would include the USA and Canada, which generally have similar standards of food and water quality and hygiene, and similar disease profiles to Europe, and therefore vaccination and other precautions may actually not be required or be thought not to be required. Travel in developing countries would presumably be more relevant to the purpose of the study and would probably be perceived to be riskier.
  2. Participation was presumably entirely self-selected, and this non-randomness could introduce some bias. People with antivaccination views are probably more likely to want to make their opinions known, compared to those with neutral or positive attitudes. The same could apply to attitudes to other issues.

Some English corrections required. Suggest authors ask a native English-speaking science writer to revise the ms.

Title punctuation should be changed to: Travelers’ Attitudes, etc. The apostrophe is obligatory.

Line 36: ‘trauma’, rather than ‘traumas’.

Line 70: unconventional use of ‘comprehending’.

Lines 74, 75: Use past tense, not future.

 Line 110: the word beware cannot be used in this way. Suggest ‘About 15.9% would not be wary of raw fruits etc…’